# A study of association between maternal tetanus toxoid immunization and neonatal mortality in the context of Bangladesh

Sujan Kumar Naha[1]☯, Md. Efty Islam Arpon[1]☯, Rifa Tasfia Siddique[1], Farjana Rahman Ripa[1], Mohammad Nayeem Hasan[1], Md. Jamal Uddin[1,2]*

1 Department of Statistics, Shahjalal University of Science & Technology, Sylhet, Bangladesh, 2 Department of Graduate Studies, Daffodil International University, Dhaka, Bangladesh

☯ These authors contributed equally to this work.
* jamal-sta@sust.edu

**Data Availability Statement:** All data files are available from the figshare database. DOI: (https://doi.org/10.6084/m9.figshare.27210603).

## Abstract

### Background

Maternal tetanus toxoid (MTT) vaccination during pregnancy remains an important factor for reducing infant mortality globally, especially in developing nations, including Bangladesh. Despite commendable progress in reducing child mortality through widespread MTT vaccination during pregnancy, the issue still exists. This analysis explores the impact of MTT vaccination on neonatal mortality in Bangladesh and identifies associated factors.

### Methods

This research utilizes data from the 2019 Bangladesh Multiple Indicator Cluster Survey (MICS). The dataset consists of 23,402 cases; among them, 587 cases resulted in infant death. The outcome variable was infant mortality, which was binary. The independent variables identified as potential contributors to the cause of death included tetanus toxoid vaccination status, mode of delivery (cesarean section or not), and mother's education level, among others. The Poisson model was employed to analyze the data.

### Results

The analyses showed that the neonatal mortality rate was 2.51%. Notably, 45.90% of mothers received the MTT vaccination during pregnancy. Among them, 23.07% received a single dose, and 22.82% took adequate doses (receiving more than two doses) and adhered to WHO guidelines. The adjusted incidence rate ratio (IRR) was 1.36, which indicates that there was a 36% higher risk of neonatal mortality for those children whose mothers did not take TT (IRR = 1.36, p = 0.081). We also found that women from middle-class households (IRR = 1.58, 95% CI = 0.98, 2.54) and women with higher parity (IRR = 1.96, 95% CI = 0.95, 4.03) also had a higher risk of newborn fatalities. A comparable trend has been observed regarding the correlation between the number of tetanus doses administered and neonatal mortality, where it also emphasizes the importance of receiving adequate doses (a minimum

**Funding:** The author(s) received no specific funding for this work.

**Competing interests:** No competing interests exist for this manuscript.

of 2 doses of tetanus vaccine) to mitigate neonatal mortality (adjusted IRR = 0.54, 95% CI = 0.29, 1.01) in comparison to no doses received.

## Conclusion

Administering a minimum of one maternal tetanus dose significantly lowers the risk of neonatal mortality. Other than Maternal Tetanus Toxoid vaccination, the analyses underscore various contributors to neonatal mortality, encompassing maternal healthcare, delivery procedures, socio-economic status, and education. Targeted interventions addressing these factors have the potential to efficiently decrease neonatal mortality rates and improve overall maternal and child health.

## Introduction

Tetanus is an acute infectious disease that continues to pose a significant public health challenge. It occurs when the umbilical stump is covered with contaminated material or when non-sterile tools are used to cut the umbilical cord [1]. It can complicate surgery, intramuscular injections, gangrene, burns, ulcers, necrotic snakebites, middle ear infections, septic abortions, and childbirth [2]. Tetanus can impact individuals across all age groups; however, it presents a particularly significant risk to newborns and pregnant women lacking adequate tetanus-toxoid vaccinations. In such cases, the disease can often result in death or significant health complications [3].

The disease is caused by *Clostridium tetani* bacteria. This common bacterium is typically found in soil, but it can also be found in human and domestic animal feces [4]. Spores of *Clostridium tetani* enter the human body through wounds or small abrasions when anaerobic conditions are met. Tetanus can be avoided with proper care whenever a wound arises and adequate vaccination [5]. However, it should be noted that tetanus cannot be completely eliminated due to the presence of spores in the natural environment.

Maternal and neonatal tetanus (MNT) is a significant public health issue that impacts mothers and their newborns. It is prevalent when women give birth in unsanitary conditions and do not receive the complete series of tetanus toxoid (TT) vaccinations. So, an effective approach to reducing the spread of the disease is through the immunization of newborns and the women who are planning to become pregnant. This immunization can be achieved by vaccination with tetanus-toxoid-containing vaccines (TTCV). Maternal and neonatal tetanus (MNT) protection through vaccination ensures that a baby remains protected from the disease during the initial two months of life [6].

The tetanus-toxoid-containing vaccines (TTCV) are typically given to reproductive women between the ages of 15 and 44 years to protect both mother and newborn and prevent maternal and neonatal tetanus. A woman needs five tetanus toxoid doses to be protected against tetanus for life [6]. For children, the World Health Organization (WHO) recommends six doses of TTCV, three of which are primary and three of which are boosters. The three-dose primary series should start as early as six weeks of age, with the following doses spaced at least four weeks apart. Preferably, the three booster doses should be administered between the ages of 12 and 23 months, 4 and 7 years, and 9 and 15 years [7]. Booster immunization campaigns are directed toward women of childbearing age, and in many nations, they have significantly decreased the incidence of tetanus in both mothers and newborns [5].

It has been challenging to enhance neonatal health in low-income countries because babies have less access to health care and are thus more at risk of worse health outcomes [8]. For instance, a prior study between 1997 and 2002 discovered that 54.7% of infant deaths happened during the neonatal period [9]. A failure in routine prenatal immunization programs, below average antenatal care services, an increase in home deliveries, and unhygienic deliveries result in a mortality rate of about 35% and proximal (309000) deaths from maternal or neonatal tetanus [10]. Tetanus caused an estimated 787,000 infant fatalities in 1988, with an estimated proportionate mortality rate of 6.7 per thousand live births, demonstrating the disease's significant contribution to the world's high rate of neonatal mortality [11]. Globally, reported newborn tetanus mortality dropped by 85% between 2000 and 2018, from 170,829 to 25,000 deaths [12]. Approximately 25,000 fewer newborns died in 2018 compared to 215,000 in 1999, a considerable decline since that year. Out of 59 low- and middle-income at-risk nations, MNT has currently been eradicated in 47 of them [13].

The Bangladesh Demographic and Health Survey reported neonatal mortality rates of 28 per 1,000 live births in 2014 and 30 in 2017, while under-five mortality rates were 46 per 1,000 in 2014 and 45 in 2017 [14, 15]. The current rate is significantly higher than the global average of 6.6 per 1,000 live births in 2015 and exceeds the 7.4 per 1,000 averages in developing countries and neighboring countries like India and Nepal [16]. The WHO reports that preterm birth causes 30% of global neonatal deaths, sepsis or pneumonia 27%, birth asphyxia 23%, congenital abnormalities 6%, neonatal tetanus 4%, diarrhea 3%, and other causes 7% [17–19]. An exploratory study in rural Bangladesh found that tetanus accounts for 42% of newborn deaths [20]. According to another study, out of 330 infant deaths, 112 of them fit the tetanus case description [21]. A study in 2016 found that there were 24.4 neonatal deaths for every 1000 live births in Bangladesh, based on 6748 neonatal fatalities spread throughout four districts (Thakurgaon, Jamalpur, Moulvibazar, and Narail), with a combined population of 6.7 million [22].

Tetanus is still prevalent and a major cause of death in low-income countries, despite considerable World Health Organization programs that effectively target maternal and newborn disease [23]. As opposed to low-income countries (LICs), High-Income Countries (HICs) like the United States of America, England, and Denmark have long since managed tetanus-related newborn mortality [24]. A few Low and Middle-Income Countries (LMICs), like Bangladesh, Afghanistan, and Nepal, are also making progress in lowering neonatal tetanus fatality rates [11] even though the pace of reduction is very low [14].

Effective intervention programs have been implemented in these countries, which include improved cord care, increased tetanus toxoid coverage for pregnant women, adoption of safe birth practices, and implementation of postpartum care procedures. Despite some successes in countries such as Bangladesh and Nepal, low- and middle-income countries (LMICs) still face challenges in reducing infant mortality caused by neonatal tetanus. According to our knowledge, no study ever before was able to give a clear picture of the influence of maternal tetanus toxoid vaccination in reducing neonatal mortality in Bangladesh. The objective of our study was to investigate the potential link between neonatal mortality and tetanus toxoid immunization among women aged 15–49 in Bangladesh. This study will assist policymakers and planners in making more informed and effective decisions regarding this important issue.

## Materials and methods

### Description of the dataset

The research was conducted using a cross-sectional study design, utilizing the accessible data from the Bangladesh Multiple Indicator Cluster Survey (MICS) conducted by the Bangladesh

Bureau of Statistics (BBS) in collaboration with the United Nations International Children's Fund (UNICEF) in 2019. The survey was carried out from January 19 to June 1, 2019. The survey sample was selected through a two-stage, stratified cluster sampling method in order to recruit the participants. The 2011 Bangladesh Census of Population and Housing was used as the sampling frame for this study. The enumeration areas (EAs) selected for the census enumeration were the primary sampling units (PSUs) selected during the initial stage. In each sample, a household listing was completed as part of the research process. In the second stage, a sample of households was selected for further analysis. The sample was created to generate estimates for various variables concerning women and children across the entire country, including both urban and rural areas in all 64 districts of the seven administrative divisions. The survey consisted of a sample of 64,400 households and approximately 3,220 primary sampling units (PSUs) [25]. The households were thoroughly interviewed by trained interviewers using questionnaires that covered reproductive history, prenatal and postnatal care, family planning, and demographic characteristics. The datasets were made available to the public for access and were specifically collected from sampling units (PSUs) [26].

Children who have passed away during the neonatal period or within the first month after birth may be considered as eligible cases for participation in the study. Moreover, it is important to note that the study subjects were specifically chosen to include children who had a high risk of newborn mortality, even though they were still alive at the time of the survey. Births that occurred prior to 2014 or five years before the survey were not classified as cases.

**Study variables.** The outcome variable is the child's survival status in the neonatal stage, which is binary and classified as either 0 (for alive) or 1 (for death). The analysis utilized data on the age at which mortality events occurred among live births that took place in the five-year period prior to the 2019 MICS. The variable indicating whether or not a neonatal death occurred was recoded using the standard procedures outlined in the SPSS syntax files provided by UNICEF [27]. These recoded data were used to create the outcome variable for the research analysis.

The principal factor under investigation as a potential exposure variable was the administration of the tetanus toxoid vaccine, classified as "0" for none-taken or "1" for taking at least 1 dose. Again, at least two doses are considered adequate according to the WHO. In addition to the principle factor, the additional modifiable risk factors analyzed in this research consisted of cesarean section (yes, no), gender of the neonate (boy and girl), antenatal care visits (less than 4, 4 and above), division (Barisal, Chittagong, Dhaka, Khulna, Rajshahi, Rangpur, and Sylhet), mother's education level (primary, higher secondary), birth order (1, 2, and 3), mother's age during pregnancy (15–19, 20–24, 25–29, 30–34, 35–39, 40–44, and 45–49), socioeconomic status (poorest, middle, and richest).

## Statistical analysis

Chi-square tests have been implemented to assess the relationship between potential exposure or control variables and infant mortality, forming the foundation for our models. Factors in bivariate models with p-value < 0.2 were included in the multivariable model, along with a previous multicollinearity test, and this threshold for p-value has been taken from a previously published paper [28, 29].

For highly skewed binary data with significant class imbalance, Poisson regression is preferred to logistic regression. Poisson regression performs better because of its log transformation, which improves results stability and interpretability. This approach is particularly advantageous in cases of high skewness, as it offers reliable estimates and reduces the risk of overfitting. The Poisson model effectively estimates relative risk and manages variance

overestimation, making it a robust choice for binary outcomes in such contexts [30, 31]. Our data's extreme imbalance and skewness align well with Poisson regression's strengths. Therefore, the final analysis was carried out using a Poisson regression model to examine the relationship between the outcome variable and several predictor variables, since in our data the proportion of death is less than 5% (approximately 2.54%), indicating a high skewness in child mortality. The model included all variables deemed significant by chi-square analysis, ensuring a comprehensive examination of their impact on child mortality. The missing values were removed with case-wise deletion as per the default data handling procedure in STATA. For each predictor variable, the incidence rate ratio (IRR) and the 95% confidence interval (CI) for child deaths were calculated.

### Ethics statement

The analysis utilized publicly accessible datasets from health surveys, ensuring that all identifiable personal information was removed. The analysis conducted in this study utilized secondary data sources, thereby exempting it from the need for ethical review approval from the relevant institutions. Nonetheless, the methods employed in MICS underwent a thorough review and received approval from both UNICEF and BBS.

### Results

The most recent MICS (2019) survey data were integrated into this study. Among the 23,402 neonatal births, there were 587 neonatal fatalities, accounting for approximately 2.54% of the neonatal deaths (Fig 1). Approximately 45.9% of women received at least one dose of tetanus toxoid immunization during their most recent pregnancy, while 22.8% of women received the adequate number of doses. A multitude of factors played a role in the occurrence of early newborn deaths. In the univariate analysis presented in Table 1, it was observed that 67.2% of women underwent cesarean section, while the percentage of normal deliveries stood at 32.8%. Notably, the death rate associated with cesarean sections was low, recorded at 1.7%. Observations indicate that neonatal mortality rates are elevated in boys at 2.7%, compared to 2.3% in

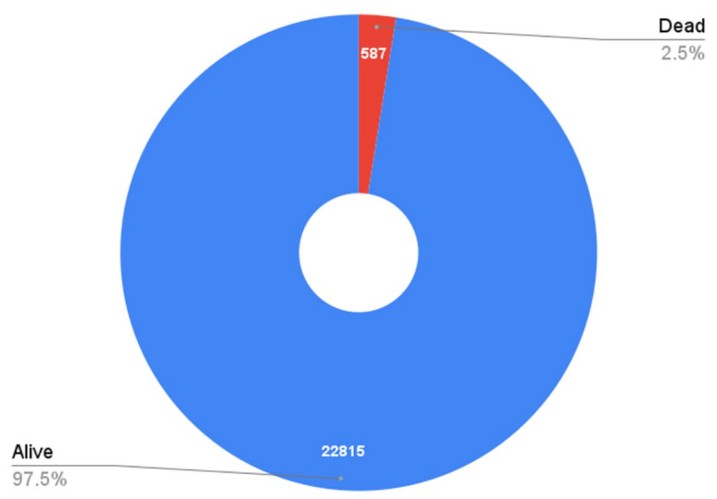

Proportion of dead and alive neonates

**Fig 1. Proportion of dead and alive neonates.**

**Table 1. The row-wise proportional distribution across various categories.**

| | Neonatal mortality status | | Total n (%) | P-value |
|---|---|---|---|---|
| | Dead n (%) | Alive n (%) | | |
| **TT vaccinated** | | | | |
| No | 202(3.3) | 5934(96.7) | 6136(54.1) | 0.255 |
| Yes | 152(2.9) | 5055(97.1) | 5207(45.9) | |
| **TT doses taken** | | | | |
| None | 202(3.3) | 5934(96.7) | 6136(54.1) | 0.254 |
| 1 injection | 84(3.2) | 2532(96.8) | 2616(23.1) | |
| 2 injections or more (adequate doses) | 68(2.6) | 2519(97.4) | 2587(22.8) | |
| **Cesarean section** | | | | |
| Yes | 117(3.1) | 3687(96.9) | 3804(67.2) | <0.001 |
| No | 90(4.8) | 1769(95.2) | 1859(32.8) | |
| **Gender** | | | | |
| Boy | 330(2.7) | 11776(97.3) | 12106(51.7) | 0.028 |
| Girl | 257(2.3) | 11039(97.7) | 11296(48.3) | |
| **Antenatal care (ANC) visits** | | | | |
| Less than 4 | 154(3.0) | 5053(97.0) | 5207(57.1) | 0.077 |
| 4 and above | 110(3.0) | 3591(97.0) | 3701(40.6) | |
| Don't know | 12(5.7) | 200(94.3) | 212(2.3) | |
| **Division** | | | | |
| Barisal | 49(2.4) | 2020(97.6) | 2069(8.8) | 0.004 |
| Chittagong | 119(2.4) | 4778(97.6) | 4897(20.9) | |
| Dhaka | 91(2.0) | 4508(98.0) | 4599(19.7) | |
| Khulna | 70(2.2) | 3158(97.8) | 3228(13.8) | |
| Mymensingh | 37(2.7) | 1341(97.3) | 1378(5.9) | |
| Rajshahi | 71(2.9) | 2379(97.1) | 2450(10.5) | |
| Rangpur | 76(2.7) | 2689(97.3) | 2765(11.8) | |
| Sylhet | 74(3.7) | 1942(96.3) | 2016(8.6) | |
| **Women education level** | | | | |
| Primary and secondary | 527(2.7) | 19236(97.3) | 19763(84.5) | 0.001 |
| Higher secondary | 60(1.6) | 3579(98.4) | 3639(15.6) | |
| **Birth order** | | | | |
| 1st parity | 240(2.8) | 8248(97.2) | 8488(36.3) | |
| 2nd parity | 264(2.2) | 11944(97.8) | 12208(52.2) | 0.002 |
| 3rd parity or more | 72(3.1) | 2459(96.9) | 2531(10.8) | |
| **Women age** | | | | |
| 15–19 | 50(2.9) | 1678(97.1) | 1728(7.4) | |
| 20–24 | 202(2.9) | 6868(97.1) | 7070(30.2) | 0.055 |
| 25–29 | 170(2.5) | 6711(97.5) | 6881(29.4) | |
| 30–34 | 91(1.9) | 4707(98.1) | 4798(20.5) | |
| 35–39 | 55(2.5) | 2188(97.5) | 2243(9.6) | |
| 40–44 | 16(2.9) | 535(97.1) | 551(2.4) | |
| 45–49 | 3(2.3) | 128(97.7) | 131(0.6) | |
| **Wealth index** | | | | |
| Poorest | 306(2.9) | 10355(97.1) | 10661(45.6) | <0.001 |
| Middle | 214(2.4) | 8613(97.6) | 8827(37.7) | |
| Richest | 67(1.7) | 3847(98.3) | 3914(16.7) | |

girls during delivery. It is worth mentioning that the neonatal mortality rates were 1.7% for mothers who had fewer than four antenatal care (ANC) visits. On the other hand, mothers who had four or more ANC visits had a slightly lower neonatal mortality rate of 1.21%. Furthermore, a minor percentage of mothers (0.13%) reported uncertainty regarding the number of antenatal care visits they had attended. Notably, a substantial correlation in univariate analysis has been identified between division and neonatal mortality, with mothers from the Sylhet division experiencing the highest neonatal mortality rate at 3.7%. It is important to highlight that education significantly influences neonatal mortality rates. Women with primary or secondary education experienced a 2.7% rate of neonate deaths, while those with higher secondary education had a rate of 1.6%. Birth order appears to be an important factor, as second-parity births are linked to increased neonatal mortality, highlighting the need for further investigation into the underlying causes. Women with increased birth parity experienced elevated neonatal mortality, which constitutes another critical factor. The wealth index of the respondents indicates that 45.6% came from poor families, 37.2% were from the middle class, and 16.7% belonged to the rich category. The neonatal mortality rates were observed to be 2.9% for the poorest, 2.4% for the middle class, and 1.7% for the richest segments of the population. Finally, it is noteworthy that a considerable proportion of women aged 20–24 experienced a significant rate of newborn infant mortality, recorded at 2.9%.

Table 1 displays the mortality status row proportions across different covariate categories. It is evident from the table that the respondents classified in the lowest wealth index category experienced the highest percentage of neonatal deaths at 2.9%, in contrast to the richest group at 1.7% and the middle class at 2.4%. The Sylhet division exhibited the highest proportion of neonatal mortality at 3.7%, closely followed by the Rajshahi division at 2.9%. The mortality rate is notably elevated in individuals with a third parity or higher, recorded at 3.1%. Additionally, mothers who experienced natural births exhibit a greater percentage of neonatal deaths (4.8%) in contrast to those who underwent cesarean sections (3.1%). Finally, the mothers who received the tetanus toxoid vaccine exhibited a lower neonatal mortality rate of 2.9% compared to the 3.3% observed in non-vaccinated mothers during their last pregnancy.

From the figures, we get an insight into the association and distribution among some variates. The second figure Fig 2 highlights the importance of tetanus vaccination in reducing neonatal deaths. Regions with higher vaccination rates among neonates, such as Barishal, show lower percentages of neonatal deaths. Conversely, areas like Sylhet, which have lower immunization rates, have a greater incidence of infant fatalities. This pattern underscores the critical role of tetanus vaccination in improving neonatal survival rates, as evidenced by the significantly lower mortality rates in areas with better vaccination coverage. Fig 3 suggests that first-parity births have the highest rate of vaccination, and with subsequent births, the proportion of vaccinated individuals decreases. Relating this to Table 1, it is observed that the 3rd or higher parity births also have a higher proportion of neonatal deaths, implying the importance of vaccination. Fig 4 represents the proportion of doses of the tetanus toxoid vaccine taken by the mother where the newborn faced a neonatal death. Further implying the significance of adequate tetanus toxoid vaccination, it is observed that, among the neonatal deaths, a significantly large proportion of mothers (56.8%) had taken no tetanus toxoid vaccine during pregnancy, and only 19% of the mothers had an adequate amount of at least 2 doses of tetanus toxoid vaccine.

The results of the regression analysis of the factors that predict newborn mortality are shown in Table 2. In the Poisson regression model, including the variables with $p < 0.2$, $p < 0.1$, $p < 0.05$, $p < 0.01$, $p < 0.001$ in the univariate analysis, tetanus doses, women's education level, women with a higher index wealth quintile, and women with a higher parity were found to be significantly linked with newborn mortality in the Poisson regression model. Crude

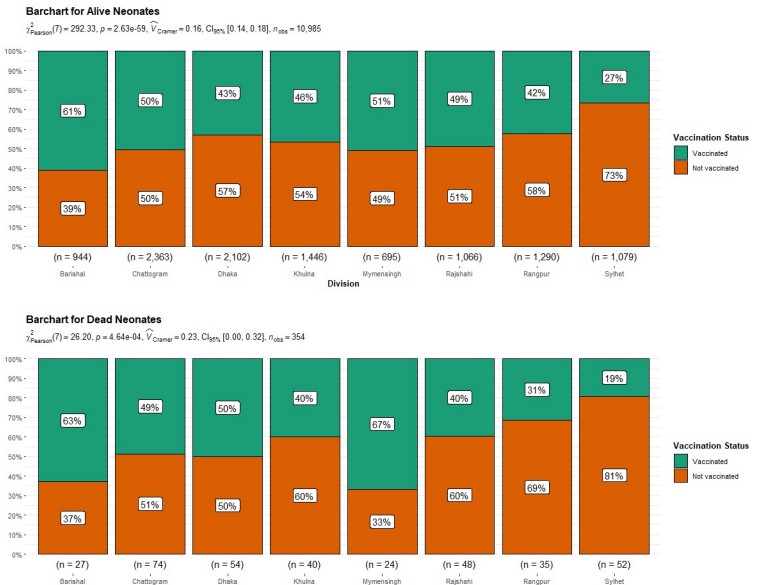

**Fig 2. Tetanus toxoid vaccination status across different divisions.**

incidence rate ratios (IRRs) for death are shown in the characteristics table for women who had tetanus toxoid (TT) immunization vs. those who did not. When compared to the vaccinated group, the unvaccinated group's crude IRR for death was 1.19 at first, with a p-value of 0.18, indicating no discernible difference in mortality. However, after adjusting for the other factors in the model, the crude IRR rose to 1.36 with a p-value of 0.08. Although this is not statistically significant at the traditional cutoff point of p < 0.05, this adjusted IRR suggests an elevated risk of MTT on infant mortality. According to our analysis, mothers with no tetanus dose had a 36% increased incidence rate ratio of experiencing newborn mortality (IRR = 1.36, 95%CI = 0.96, 1.93) compared to mothers who took at least a single tetanus dose. A similar trend has been found in the relationship between the number of tetanus doses taken and neonatal mortality that has been given in S1 Table, which also describes taking adequate doses (at

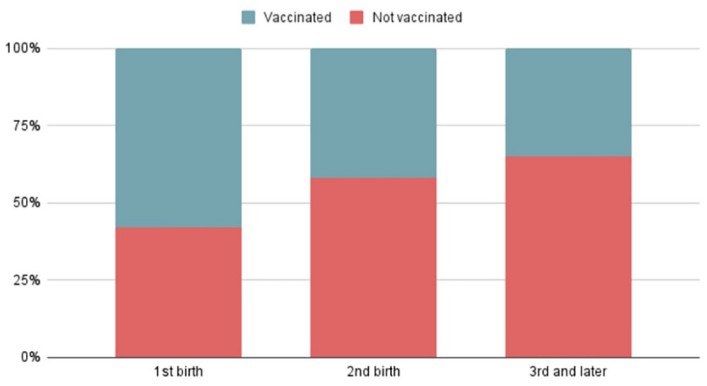

Tetanus toxoid vaccination status among different birth order

**Fig 3. Tetanus toxoid vaccination status among different birth order.**

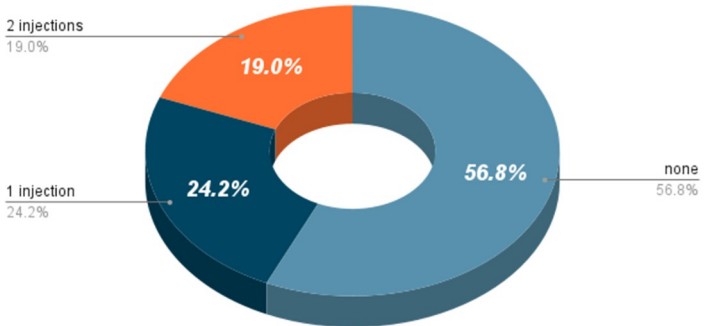

Tetanus Toxoid Vaccination Among Mothers with Neonatal Mortality

**Fig 4. Tetanus toxoid vaccination among mothers with neonatal mortality.**

least 2 doses of tetanus vaccine) to reduce neonatal mortality (Crude IRR = 0.87, 95% CI = 0.57, 1.32) compared to taking none. Interestingly, the cesarean section was found to be insignificant. We also discovered that women's education level was significantly associated with newborn mortality, mothers with primary or secondary education had an increased 86% incidence rate ratio of newborn mortality (IRR = 1.86, 95%CI = 1.16, 2.99) compared to mothers who had studied at the higher secondary level. Additionally, women from poor families had a significantly higher incident rate ratio compared to the women from rich families (IRR = 1.77, 95%CI = 0.93, 3.38). Similarly, women from middle-class families had a significantly higher incident rate ratio compared to the women from rich families (IRR = 1.58, 95% CI = 0.98, 2.54). In addition, third-parity births were found to be substantially correlated with newborn mortality when compared to the second-parity births (IRR = 1.96, 95%CI = 0.95, 4.03). This indicates that women with higher parity had a 96% increased risk of neonatal deaths. Women aged between 25 and 29 years indicated a fairly significant association (IRR = 1.89, p = 0.081) with newborn death, showing an 89% higher incident rate ratio of neonatal mortality compared to women aged between 30 and 35 years. Likewise, women aged between 45 and 49 years had a significantly (IRR = 4.05, p = 0.055) higher rate ratio than women aged between 30 and 35 years. Gender, antenatal care visits, and division, which are the necessary factors, were found to be insignificant in the model.

## Discussion

Tetanus demonstrates a significant threat to human life [32]. Despite significant advancements over the past twenty years in reducing global tetanus-related fatalities, neonatal tetanus has garnered considerable focus, while maternal tetanus remains largely underexplored [24, 33]. This survey aims to identify the impact of neonatal mortality in relation to tetanus toxoid vaccination among pregnant women aged 15 to 49 years. The findings indicate that maternal vaccination plays a significant role in decreasing neonatal death rates.

Previous investigations have not explored the relationship between neonatal mortality and tetanus toxoid immunization in conjunction with other essential covariates. Our primary results demonstrate that women who obtain at least one dose of the tetanus vaccine substantially reduce the likelihood of neonatal death in their children. Furthermore, around 2.54% of neonatal deaths in Bangladesh are associated with pregnant women who did not receive at least one dose of the tetanus toxoid vaccination. In contrast, about 45.9% of women have

**Table 2. Factor associated with neonatal mortality.**

| Characteristics | Crude IRR (95% CI) | p-value | Adjusted IRR (95% CI) | p-value |
|---|---|---|---|---|
| **TT vaccinated** | | | | |
| Yes | 1 | | 1 | |
| No | 1.19(0.92,1.52) | 0.181 | 1.36(0.96,1.93) | 0.081 |
| **Cesarean section** | | | | |
| Yes | 1 | | 1 | |
| No | 1.48(1.04,2.10) | 0.028 | 1.32(0.93,1.88) | 0.123 |
| **Gender** | | | | |
| Boy | 1.23(1.02,1.47) | 0.030 | 1.27(0.92,1.75) | 0.151 |
| Girl | 1 | | 1 | |
| **ANC visits** | | | | |
| Less than 4 | 1 | | 1 | |
| Above 4 | 1.10(0.80,1.50) | 0.556 | 1.20(0.79,1.83) | 0.395 |
| Don't know | 1.22(0.60,2.46) | 0.578 | 1.51(0.68,3.39) | 0.314 |
| **Division** | | | | |
| Barisal | 0.81(0.54,1.22) | 0.309 | 1.18(0.57,2.44) | 0.659 |
| Chittagong | 0.88(0.63,1.22) | 0.428 | 1.13(0.58,2.20) | 0.729 |
| Dhaka | 0.76(0.54,1.08) | 0.123 | 1.31(0.70,2.45) | 0.392 |
| Khulna | 0.86(0.58,1.27) | 0.446 | 1.05(0.54,2.05) | 0.884 |
| Mymensingh | 0.90(0.59,1.37) | 0.612 | 1.28(0.51,3.24) | 0.599 |
| Rajshahi | 1.03(0.72,1.48) | 0.854 | 1.25(0.63,2.49) | 0.526 |
| Sylhet | 1.45(0.97,2.16) | 0.071 | 1.66(0.65,4.25) | 0.289 |
| Rangpur | 1 | | 1 | |
| **Women education level** | | | | |
| Primary or secondary | 1.73(1.27,2.34) | <0.001 | 1.86(1.16,2.99) | 0.011 |
| Higher secondary | 1 | | 1 | |
| **Birth order** | | | | |
| 1st parity | 1.29(1.07,1.56) | 0.007 | 1.11(0.75,1.64) | 0.611 |
| 2nd parity | 1 | | 1 | |
| 3rd parity | 1.42(1.08,1.88) | 0.014 | 1.96(0.95,4.03) | 0.067 |
| **Women Age** | | | | |
| 15–19 | 1.61(1.09,2.37) | 0.017 | 1.36(0.60,3.09) | 0.461 |
| 20–24 | 1.68(1.25,2.27) | 0.001 | 1.89(0.93,3.86) | 0.081 |
| 25–29 | 1.50(1.11,2.02) | 0.009 | 1.62(0.86,3.04) | 0.134 |
| 30–34 | 1 | | 1 | |
| 35–39 | 1.34(0.90,1.99) | 0.143 | 1.09(0.46,2.54) | 0.851 |
| 40–44 | 1.76(0.95,3.25) | 0.073 | 2.35(0.44,12.45) | 0.317 |
| 45–49 | 1.95(0.63,6.05) | 0.247 | 4.05(0.97,16.95) | 0.055 |
| **Wealth index** | | | | |
| Poorest | 1.70(1.25,2.33) | 0.001 | 1.77(0.93,3.38) | 0.085 |
| Middle | 1.39(1.01,1.90) | 0.041 | 1.58(0.98,2.54) | 0.060 |
| Richest | 1 | | 1 | |

* IRR = Incidence Rate Ratio

received at least one dose of the vaccine, which could serve as valuable data for initiatives aimed at enhancing vaccination rates.

Our findings show that only 22.82% of pregnant women had received a sufficient amount of the TT vaccine, which is low as compared to a study showing that sufficient TT immunization among pregnant women was 75% worldwide [34]. These findings are consistent with previous research's findings, which showed a lower prevalence of obtaining the adequate TT vaccine [21] and the rate of newborn death was 63.8 per 1000 live births [35]. The prevalence of TT immunization was 81.8% for MICS (2006) and 61.3% for MICS (2012–2013) [36] and we observed in our research that for MICS (2019), the rate of taking tetanus doses was 54.1%. These findings shed light on the varying levels of vaccination coverage among pregnant women, emphasizing the need to promote and ensure full compliance with the WHO's recommended vaccination regimen during pregnancy for the optimal prevention of maternal and neonatal tetanus. Overall, in Bangladesh, the prevalence has generally been decreasing over the course of the survey years.

In our model, we discovered that tetanus doses taken, women's education level, wealth index, and birth order were significantly associated with higher odds of neonatal mortality. Our analysis shows that a mother with no tetanus doses has 36.4% more chance to experience neonatal death (IRR = 1.36, 95%CI = 0.96, 1.93) compared to a mother with at least 1 dose. Similar to other studies, we discovered that the TT vaccine had a protective effect against infant mortality when compared to IFA (iron–folic acid) supplementation alone [37–39]. The evidence from India aligns with our analysis, indicating that receiving at least one dose of the TT vaccination could decrease infant death [40]. Another study revealed a sharp reduction in newborn mortality as a result of a decrease in neonatal tetanus deaths [41]. Tetanus vaccination coverage plays a vital role in reducing neonatal tetanus. We are the first to show the association between tetanus toxoid vaccination and neonatal mortality in Bangladesh in recent years.

Cesarean section is a potential factor for neonatal mortality according to previous studies, where cesarean section delivery is highly correlated with neonatal mortality in low-income countries like Bangladesh [8, 42]. However, we found no association between neonatal mortality and cesarean sections, which may be a noticeable gap in our study that could be addressed with the confirmation of further research.

A mother's education significantly plays a vital role in neonatal mortality. Women who only completed elementary or secondary school had a greater risk of newborn mortality compared to women who pursued higher education. Considering that educated women may be better empowered to make decisions about their health, education may increase their understanding of the harmful impacts of tetanus and neonatal death [36, 43].

Another important factor associated with neonatal mortality is higher parity. Our study revealed that women with higher parity have higher odds of neonatal mortality compared to those with lower parity. According to our findings, women with three or more parity had 43% greater odds when compared to women with second parity. This is supported by similar findings from previous studies that have found higher parity to be a significant predictor of neonatal mortality [44–46]. The past birth experiences of the ladies may be the cause. It's also reasonable that women with more children are less likely to be employed or to have greater levels of education. There are further studies that show older women in the nation are less likely than younger women to use prenatal and delivery care, which can be another identifiable reason [47–49].

Various age groups in the reproductive age of women play a crucial role in reducing neonatal death. Older women (age 45–49 with IRR = 4.05, 95%CI = 0.97, 16.95) are more exposed to neonatal death than any other reproductive period, and it is one of the important issues that

needs to be taken care of. Similar studies show that maternal periods in different age groups directly influence neonatal mortality, which is likely due to complications of delivery that arise at a later age [50, 51].

The health of the Bangladeshi population has significantly improved over the past 20 years, and Bangladesh has been acknowledged as an example of "good health at low cost" [52]. As far as we are aware, socioeconomic status has a big impact and is highly correlated; babies from the "poorest" homes are more likely to die, which emphasizes how critical it is to solve socio-economic gaps in access to healthcare. Our research says women from wealthy households are less likely to face neonatal mortality compared to women with a poor wealth quintile index. This result is in line with earlier research demonstrated that increasing the wealth index of women living at home protects against tetanus in comparison to a low wealth index [40, 43, 53, 54]. We believe that women from wealthy families are more likely to have access to health-care services than women from low-income families. The implementation of policies and pro-grams that guarantee all pregnant women receive a minimum of one dosage of prenatal tetanus toxoid vaccination, especially those in need or marginalized population subgroups, is expected to have a significant influence on improving neonatal survival. This specific interven-tion should be within the grasp of the public health system as an immediate priority due to dif-ficulties in providing the entire health system functionality needed for comprehensive and safe mother and newborn care [36].

## Limitations and strengths

### Strengths

To our knowledge, this study is the first in Bangladesh to investigate the relationship between maternal tetanus toxoid vaccination and neonatal mortality. We employed appropriate data analysis techniques, taking into consideration all intricate survey designs. Findings from this work can be used to inform future research, policy, and clinical practice and to benchmark progress. One of the benefits of this study is the sizeable and nationally representative sample size, and the results are applicable to the entire country. The information was gathered using the most recent survey. Our results would surely pique interest in additional research and edu-cate decision-makers about the gaps in tetanus care that need to be filled.

### Limitations

However, despite the several strengths, the existence of bias resulting from different survey time points and the cross-sectional character of the data cannot be confirmed. Some variables that were statistically insignificant but still important for the research study have been consid-ered in the model. It is crucial to keep in mind that the mortality that is being discussed here includes deaths from all causes. Regretfully, our analysis lacks particular data that isolates the risk exclusively for newborn mortality. Notwithstanding this restriction, the trend that has been seen points to a higher risk of death for those who have not had the TT vaccination; how-ever, statistical significance is not attained. This realization emphasizes how crucial it is to con-duct additional studies to determine the specific effect of the TT vaccine on mortality outcomes, especially neonatal death. To offer more precise information, future research focus-ing on infant mortality rates and any confounding factors is required.

The level of significance was deemed to be fairly high. In addition, the study's drawback also emerges from the fact that we had little control over the correlated variables to include in the analysis due to the secondary data source we employed. Conclusions regarding a causal association, the relative contributions of immunization prior versus during the most recent pregnancy, or the best possible ways to increase coverage are not possible due to the cross-

sectional character of this investigation. In addition, the dependent variable for newborn death has a large number of missing values, making it impossible to include them in the analysis. This could lead to bias, and the variable varies over time, potentially changing the claimed association in longitudinal studies.

## Conclusion

Premature infant death is more common among Bangladeshi women with no tetanus toxoid vaccination. Based on our findings, taking at least one maternal tetanus dose greatly reduces neonatal mortality, which clearly shows that increasing vaccination coverage as well as deploying the mandatory law of taking at least one maternal tetanus dose can save infants in great numbers. Additional investigation is required to determine the pattern of declining infant mortality in Bangladesh following vaccination against maternal tetanus toxoid since there is no evidence related to this. Furthermore, Women with higher education levels, those in the wealthiest quintile, and those with greater parity were all significantly linked to reduced odds of newborn mortality. To decrease neonatal death for the betterment of a country like Bangladesh, we advise taking help from the government and other organizations in conducting immunization campaigns, increasing vaccination coverage, and improving self-care through proper education and awareness.

## Supporting information

**S1 Table. Analysis results for examining neonatal mortality against tetanus doses taken by mothers.**
(DOCX)

**S1 Checklist. STROBE statement—Checklist of items that should be included in reports of *cross-sectional studies*.**
(DOCX)

## Acknowledgments

We acknowledge UNICEF and the Bangladesh Bureau of Statistics for allowing us to use the data.

## Author Contributions

**Conceptualization:** Sujan Kumar Naha, Md. Efty Islam Arpon, Rifa Tasfia Siddique, Md. Jamal Uddin.

**Formal analysis:** Sujan Kumar Naha.

**Investigation:** Sujan Kumar Naha, Md. Efty Islam Arpon.

**Methodology:** Sujan Kumar Naha, Md. Efty Islam Arpon, Md. Jamal Uddin.

**Project administration:** Sujan Kumar Naha, Md. Jamal Uddin.

**Resources:** Md. Efty Islam Arpon, Rifa Tasfia Siddique, Mohammad Nayeem Hasan.

**Software:** Sujan Kumar Naha.

**Supervision:** Mohammad Nayeem Hasan, Md. Jamal Uddin.

**Validation:** Md. Efty Islam Arpon, Mohammad Nayeem Hasan, Md. Jamal Uddin.

**Visualization:** Md. Efty Islam Arpon.

**Writing – original draft:** Sujan Kumar Naha, Md. Efty Islam Arpon, Rifa Tasfia Siddique, Farjana Rahman Ripa.

**Writing – review & editing:** Sujan Kumar Naha, Md. Efty Islam Arpon, Rifa Tasfia Siddique, Farjana Rahman Ripa, Mohammad Nayeem Hasan.

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
