## [Decision Letter · Decision Letter 0]

5 Jun 2024

PONE-D-24-08202Maternal Tetanus Toxoid Immunization and Neonatal Mortality in the Context of Bangladesh: a study of association between neonatal mortality and tetanus toxoid vaccinationPLOS ONE

Dear Dr. Uddin,

Thank you for submitting your manuscript to PLOS ONE. After careful consideration, we feel that it has merit but does not fully meet PLOS ONE’s publication criteria as it currently stands. Therefore, we invite you to submit a revised version of the manuscript that addresses the points raised during the review process.**The title may be “A Study of Association Between Neonatal Mortality and Maternal Tetanus Toxoid Immunization in the Context of Bangladesh”.****In Methods of Abstract, the authors are requested to mention the sample size.****It is necessary to update the Introduction section based on the statistics and studies focusing on this topic in the context of Bangladesh.****The authors write “Factors significant in bivariate models (p < 0.2) were included in the multivariable model…”. Justify the use of p<0.2 and add references.****Add a strengths and limitations section.**

We look forward to receiving your revised manuscript.

Kind regards,

Md. Moyazzem Hossain

Academic Editor

PLOS ONE

Journal Requirements:

2. In the online submission form, you indicated that The data underlying the results presented in the study are available from (sujan03@student.sust.edu)

Reviewers' comments:

Reviewer's Responses to Questions

**Comments to the Author**

1. Is the manuscript technically sound, and do the data support the conclusions?

Reviewer #1: Partly

Reviewer #2: Partly

2. Has the statistical analysis been performed appropriately and rigorously? 

Reviewer #1: Yes

Reviewer #2: No

3. Have the authors made all data underlying the findings in their manuscript fully available?

Reviewer #1: Yes

Reviewer #2: Yes

4. Is the manuscript presented in an intelligible fashion and written in standard English?

Reviewer #1: No

Reviewer #2: No

5. Review Comments to the Author

**Reviewer #1: **I salute the authors for the great efforts put into the work.

The manuscript however has lots of errors of syntax and semantics.

I will advice the authors to take a very good look at the findings of the study and possibly modify the title of the work as it is very difficult to link maternal tetanus vaccination with all-cause neonatal mortality as they have tried to do in this study. Maternal tetanus vaccination does not have effect on several other causes of neonatal mortality, so it will be difficult to attribute maternal tetanus vaccination alone to neonatal mortality.

Some of the references were not properly cited, they should be corrected.

**Reviewer #2: **1. This is not a study finding, rather this is the findings from the existing public domain health survey datasets. I think it will be more appropriate if the manuscript title also reflect this. Authors need to change the title as “Maternal Tetanus Toxoid Immunization and Neonatal Mortality in Bangladesh: The findings from a retrospective analysis of the data from Bangladesh Multiple Indicator Cluster Survey”

2. Authors write this as study in the abstract (line 35, 49) and in many place in the manuscript. Please change the study as analysis or survey.

3. Referencing need massive correction. In discussion 32 ref came earlier than 29, 31…

4. Define ‘adequate doses’ in the method section not in result section (line 168)

5. Please reproduce the figure 1 with addition of alive and death group for all divisions within the vaccinated and non-vaccinated participants and calculate p value here.

6. Please remove figure 3. This figure has no relation with TT vaccine.

7. Massive English correction is needed for the full manuscript.

8. There is no limitation explain in the paper. Like, this is an analysis from the existing public domain health survey datasets. Prospective study may reveled more in depth information.

6. PLOS authors have the option to publish the peer review history of their article (what does this mean?). If published, this will include your full peer review and any attached files.

Reviewer #1: **Yes: **Ezra Olatunde Ogundare

Reviewer #2: No

---

## [Author Response · Author response to Decision Letter 0]

7 Aug 2024

Dear Dr Hossain,

Thank you for your invitation to submit a revised version of our manuscript "Maternal Tetanus Toxoid Immunization and Neonatal Mortality in the Context of Bangladesh: a study of association between neonatal mortality and tetanus toxoid vaccination" (PONE-D-24-08202). 

According to the comments made by the respected reviewers we modified our manuscript. Below we address all comments made by the reviewers’ point-by-point (reviewers’ comments in italic, our responses in non-italic plain font and page numbers are according to the track change version of the revised manuscript). We look forward to your positive response.

Sincerely, 

On behalf of all authors, 

Dr. Md Jamal Uddin (Corresponding authors)

Professor of Biostatistics and Epidemiology, 

Department of Statistics,

Shahjalal University of Science and Technology,

Sylhet, Bangladesh.

Phone: +8801716972846

Email: jamal-sta@sust.edu

---

## [Decision Letter · Decision Letter 1]

9 Sep 2024

PONE-D-24-08202R1A Study of Association Between Maternal Tetanus Toxoid Immunization and Neonatal Mortality in the Context of BangladeshPLOS ONE

Dear Dr. Uddin,

Thank you for submitting your manuscript to PLOS ONE. After careful consideration, we feel that it has merit but does not fully meet PLOS ONE’s publication criteria as it currently stands. Therefore, we invite you to submit a revised version of the manuscript that addresses the points raised during the review process.

We look forward to receiving your revised manuscript.

Kind regards,

Md. Moyazzem Hossain

Academic Editor

PLOS ONE

Reviewers' comments:

Reviewer's Responses to Questions

**Comments to the Author**

1. If the authors have adequately addressed your comments raised in a previous round of review and you feel that this manuscript is now acceptable for publication, you may indicate that here to bypass the “Comments to the Author” section, enter your conflict of interest statement in the “Confidential to Editor” section, and submit your "Accept" recommendation.

Reviewer #1: (No Response)

2. Is the manuscript technically sound, and do the data support the conclusions?

Reviewer #1: Partly

3. Has the statistical analysis been performed appropriately and rigorously? 

Reviewer #1: Yes

4. Have the authors made all data underlying the findings in their manuscript fully available?

Reviewer #1: Yes

5. Is the manuscript presented in an intelligible fashion and written in standard English?

Reviewer #1: No

6. Review Comments to the Author

Reviewer #1: Once again, I say well done to the authors for the job done. I still have my bias about what the manuscript is trying to prove. My reason for this bias is based on the outcome of the statistical analysis of this study which showed some factors as significant contributors to neonatal mortality. However, the authors focus on one factor that depends on those significant factors and far more important causes of neonatal mortality.

Also, the authors need to go over the manuscript as the manuscript still has some grammatical errors.

7. PLOS authors have the option to publish the peer review history of their article (what does this mean?). If published, this will include your full peer review and any attached files.

Reviewer #1: No

---

## [Author Response · Author response to Decision Letter 1]

12 Oct 2024

Thank you for your invitation to submit a revised version of our manuscript "Maternal Tetanus Toxoid Immunization and Neonatal Mortality in the Context of Bangladesh: a study of association between neonatal mortality and tetanus toxoid vaccination" (PONE-D-24-08202R1). 

According to the comments made by the respected reviewers we modified our manuscript. We look forward to your positive response.

Sincerely, 

On behalf of all authors, 

Dr. Md Jamal Uddin (Corresponding authors)

Professor of Biostatistics and Epidemiology, 

Department of Statistics,

Shahjalal University of Science and Technology,

Sylhet, Bangladesh.

Phone: +8801716972846

Email: jamal-sta@sust.edu

---

## [Decision Letter · Decision Letter 2]

11 Nov 2024

PONE-D-24-08202R2A Study of Association Between Maternal Tetanus Toxoid Immunization and Neonatal Mortality in the Context of BangladeshPLOS ONE

Dear Dr. Uddin,

Thank you for submitting your manuscript to PLOS ONE. After careful consideration, we feel that it has merit but does not fully meet PLOS ONE’s publication criteria as it currently stands. Therefore, we invite you to submit a revised version of the manuscript that addresses the points raised during the review process.

We look forward to receiving your revised manuscript.

Kind regards,

Md. Moyazzem Hossain, PhD

Academic Editor

PLOS ONE

Journal Requirements:

Reviewers' comments:

Reviewer's Responses to Questions

**Comments to the Author**

1. If the authors have adequately addressed your comments raised in a previous round of review and you feel that this manuscript is now acceptable for publication, you may indicate that here to bypass the “Comments to the Author” section, enter your conflict of interest statement in the “Confidential to Editor” section, and submit your "Accept" recommendation.

Reviewer #1: (No Response)

2. Is the manuscript technically sound, and do the data support the conclusions?

Reviewer #1: Yes

3. Has the statistical analysis been performed appropriately and rigorously? 

Reviewer #1: Yes

4. Have the authors made all data underlying the findings in their manuscript fully available?

Reviewer #1: Yes

5. Is the manuscript presented in an intelligible fashion and written in standard English?

Reviewer #1: No

6. Review Comments to the Author

Reviewer #1: I want to thank the authors for improving the manuscript. However, there are still some issues yet to be addressed or corrected. Those few issues have been highlighted in the manuscript and the comments written in those sections.

7. PLOS authors have the option to publish the peer review history of their article (what does this mean?). If published, this will include your full peer review and any attached files.

Reviewer #1: No

---

## [Author Response · Author response to Decision Letter 2]

18 Nov 2024

Comments from reviewer:

Reviewer #1

Reviewer: I believe that the authors can arrange the age in ascending order (Page 4, line no. 69).

Response: Thank you for notifying us point by point. We have re-arrange the age in ascending order.

Reviewer: This should be 15 - 19 in order to avoid overlap with the next age range 20 – 24 (page 7, line 149).

Response: We thank the editor for pointing out this one. The correction on this has been done.

Reviewer: party (page 10, line no. 208.)

Response: Thanks. This spelling mistake has been taken care of.

Reviewer: I believe that the authors inadvertently substituted the values written in this sentence. The authors need to review this sentence please (Page 10, line no. 210-212).

Response: Again, we thank the editor whole heartedly for pointing this major mistake. The values were interchanged and written as, “Finally, the mothers who received the tetanus toxoid vaccine exhibited a lower neonatal mortality rate of 2.9% compared to the 3.3% observed in non-vaccinated mothers during their last pregnancy.”

Reviewer: The bracket should be removed (page 11, line 224).

Response: Thank you. All the brackets from all figure number has been removed in line 217, 221, 224.

Reviewer: This should be corrected - compared to taking none (page 12, line no. 249).

Response: Thanks. This has been corrected.

Reviewer: Is this the true finding of this study? The authors need to review this sentence as it contradicts the next referenced sentence (page 14, line no. 310 - 311).

Response: Thank you again. This issue has been taken care of and corrected as, “Our study revealed that women with higher parity have higher odds of neonatal mortality compared to those with lower parity. According to our findings, women with three or more parity had 43% grater odds comparing to women with second parity. This is supported by similar findings from previous studies that have found higher parity to be a significant predictor of neonatal mortality (44–46).”

Reviewer: quantile or quintile? (page 15, line 327)

Response: Thank you. “quantile” was replaced with “quintile”.

Reviewer: This sentence needs to be rephrased please. (page 15, line no 330-331)

Response: We can’t thank you enough. The line “A policy and programming requirement to ensure that all pregnant women receive at least one dose of prenatal medication” is rephrased as “The implementation of policies and programs that guarantee all pregnant women receive a minimum of one dosage of prenatal medication” at line no. 331.

---

## [Decision Letter · Decision Letter 3]

18 Dec 2024

A Study of Association Between Maternal Tetanus Toxoid Immunization and Neonatal Mortality in the Context of Bangladesh

PONE-D-24-08202R3

Dear Dr. Uddin,

We’re pleased to inform you that your manuscript has been judged scientifically suitable for publication and will be formally accepted for publication once it meets all outstanding technical requirements.

Kind regards,

Md. Moyazzem Hossain, PhD

Academic Editor

PLOS ONE

Additional Editor Comments (optional):

Reviewers' comments:

Reviewer's Responses to Questions

**Comments to the Author**

1. If the authors have adequately addressed your comments raised in a previous round of review and you feel that this manuscript is now acceptable for publication, you may indicate that here to bypass the “Comments to the Author” section, enter your conflict of interest statement in the “Confidential to Editor” section, and submit your "Accept" recommendation.

Reviewer #1: All comments have been addressed

2. Is the manuscript technically sound, and do the data support the conclusions?

Reviewer #1: Yes

3. Has the statistical analysis been performed appropriately and rigorously? 

Reviewer #1: Yes

4. Have the authors made all data underlying the findings in their manuscript fully available?

Reviewer #1: Yes

5. Is the manuscript presented in an intelligible fashion and written in standard English?

Reviewer #1: No

6. Review Comments to the Author

Reviewer #1: I thank the authors for improving the manuscript and for their patience. The few comments that I have are editorial. I say well done to the authors.

7. PLOS authors have the option to publish the peer review history of their article (what does this mean?). If published, this will include your full peer review and any attached files.

Reviewer #1: No

---

## [Editor Report · Acceptance letter]

7 Jan 2025

PONE-D-24-08202R3 

PLOS ONE

Dear Dr. Uddin, 

I'm pleased to inform you that your manuscript has been deemed suitable for publication in PLOS ONE. Congratulations! Your manuscript is now being handed over to our production team.

Kind regards, 

on behalf of

Professor Md. Moyazzem Hossain 

Academic Editor

PLOS ONE